# Improving the robustness and efficiency of cell sheet protocol for breast cancer induction in animal model: A Geltrex plus Gelatin approach

Alaa T. Alshareeda[1,2]*, Nada Albarakati[3‡], Yasser Alshawakir[4], Ayidah Alghuwainem[2], Batla S. Al-Sowayan[5], Abdul Latif Khan[6], Abdullah Almubarak[4], Sarah Al-Maiman[6], Ahood Al Sayed[1], Nur Khatijah Mohd Zin[7], Amal S. Alhamid[1]

**1** Saudi Biobank Department, King Abdullah International Medical Research Centre/King Saud Bin Abdulaziz University for Health Sciences, Ministry of National Guard Health Affairs, Riyadh, Saudi Arabia, **2** Department of Blood and Cancer Research, King Abdullah International Medical Research Centre/ King Saud Bin Abdulaziz University for Health Sciences, Ministry of National Guard Health Affairs, Riyadh, Saudi Arabia, **3** Department of Blood and Cancer Research, King Abdullah International Medical Research Center/ King Saud Bin Abdulaziz University for Health Sciences, Ministry of National Guard Health Affairs, Jeddah, Saudi Arabia, **4** College of Medicine, Experimental Surgery and Animal Laboratory, King Saud University, Riyadh, Saudi Arabia, **5** National Livestock and Fishery Development Program, Ministry of Environment, Water and Agriculture, Riyadh, Saudi Arabia, **6** Department of Pathology and Laboratory Medicine, King Abdulaziz Medical City, National Guard Health Affairs, Riyadh, Saudi Arabia, **7** Faculty of Forestry and Environmental Studies, University Putra Malaysia, Selangor, Malaysia

‡ First joint.
* alshareedaa@kaimrc.edu.sa

## Abstract

### Background

Breast cancer remains a global health challenge, necessitating improved preclinical models that better mimic the complexity of the disease. The cell sheet technique holds promise in creating three-dimensional tissue constructs resembling breast cancer tissue. However, maintaining cell sheet integrity during transplantation poses challenges, particularly with low junction cells. This study aims to establish a modified protocol utilizing Geltrex™ and Gelatin to fabricate cell sheets from MDA-MB-231 cells and evaluate their tumorigenic potential *in vivo*.

### Method

We developed a novel protocol for fabricating cell sheets from MDA-MB-231 cells using Geltrex™ and Gelatin. This construct was then used to induce breast cancer *in vivo*. The novel protocol was compared to the conventional cell injection method by monitoring tumor progression *in vivo*.

**Data availability statement:** All relevant data are within the manuscript.

**Funding:** This research was funded by King Abdullah International Medical Research Center (KAIMRC), Riyadh, Saudi Arabia; grant number "RC15/136R". The funders had no role in study design, data collection and analysis, decision to publish, or preparation of the manuscript.

**Competing interests:** The authors have declared that no competing interests exist.

## Results

The novel protocol enhanced cell sheet transplant efficiency by providing scaffold support and temporary adhesion. It successfully induced breast cancer *in vivo* and facilitated metastasis, closely mimicking the progression of human breast cancer.

## Conclusion

This study highlights the potential of Geltrex™ and Gelatin as carriers for poor junction cell sheets. This original research study introduces a methodological advance in cell sheet fabrication, combining preclinical validation with technical innovation to address challenges in modeling triple-negative breast cancer (TNBC). By offering a more accurate *in vivo* representation of tumor development, this protocol enhances our understanding of breast cancer biology. The practical implications are promising, as this research can lead to more effective methods for generating *in vivo* models and tissue-engineered constructs for cancer research and regenerative medicine.

## 1. Introduction

Breast cancer (BC) is a heterogeneous disease and one of the most common malignancies affecting women worldwide, with an estimated 2.3 million new cases diagnosed annually [1]. Despite significant advancements in understanding BC, the lack of reliable preclinical models continues to hinder the development of effective therapies. BC remains a leading cause of cancer-related mortality among women globally, highlighting the urgent need for in vitro models that more accurately mimic the tumor microenvironment. Current models, such as cell line xenografts and genetically engineered mouse models, have substantial limitations in simulating the diverse aspects of human breast tissue. These limitations underscore the potential benefits of tissue engineering approaches, which could offer more accurate and representative preclinical models.

Cell suspension injection is a commonly used method for developing BC animal models. This approach involves injecting cancer cells subcutaneously or directly into the mammary fat pad of mice [2]. While this method offers advantages, such as ease of use and control over the number of cells injected, it also presents notable disadvantages [2,3]. A significant limitation is that injecting a large number of cells can lead to rapid tumor formation, which does not accurately reflect the slow progression of human BC. Furthermore, the microenvironment of the mammary fat pad, including the extracellular matrix and immune cells, may differ from that of human breast tissue, influencing the behavior and response of the injected cells [4].

Other methods for developing BC animal models, such as xenografts, also have limitations, including poor reproducibility and the potential for immune rejection [5]. These challenges further emphasize the need for innovative tissue engineering approaches that can create more physiologically relevant conditions. By better mimicking the human breast tissue environment, these approaches may lead to improved understanding and treatment of BC.

The cell sheet technique has emerged as a promising tool in tissue engineering and regenerative medicine [6,7]. This technique utilizes temperature-responsive culture dishes that allow for the detachment of cells as a sheet by reducing the culture temperature. The resulting cell sheet generates a three-dimensional (3D) tissue construct that mimics the architecture and functionality of native tissue [8]. This method has been adapted for cancer research to develop both *in vitro* and *in vivo* cancer models [9]. Notably, the cell sheet technique offers advantages over current preclinical models, including enhanced cell-cell and cell-matrix interactions and the ability to create 3D constructs that better replicate the complexity of the cancer microenvironment. Additionally, utilizing human-derived cell lines in this model increases its relevance to human disease and may facilitate the development of personalized cancer therapies [8,10,11]. However, challenges remain, such as the formation of poor junctions in some cell types [12]. These poor junctions can lead to decreased mechanical strength, reduced functionality, and an increased risk of detachment or fragmentation upon transplantation, compromising the effectiveness of the cell sheet in repairing damaged tissue or modeling disease.

This study focuses on the MDA-MB-231 cell line, a widely used model for studying BC both *in vitro* and *in vivo* [13,14]. This cell line is characterized by poor junctional complexes, which can significantly affect cellular behavior and responses to various stimuli [12]. Junctional complexes are critical for maintaining the integrity of epithelial tissues, regulating cell-cell adhesion, polarity, and signaling [15]. In BC, the loss or dysfunction of these complexes is a hallmark of the epithelial-to-mesenchymal transition (EMT), contributing to invasiveness, metastasis, and resistance to therapy [12]. The poor formation of junctional complexes in MDA-MB-231 cells has implications for research outcomes, potentially affecting drug responses and interactions with the tumor microenvironment. It also impacts the accuracy of *in vitro* assays designed to mimic *in vivo* conditions, such as Transwell migration and invasion assays, which rely on intact junctional complexes for accurate modeling of cell behavior [16].

To address these challenges, hydrogels, such as Geltrex™, can be utilized. Hydrogels are water-swollen polymer networks that mimic the properties of natural extracellular matrices (ECMs) found in living tissue [17]. Geltrex™ is particularly suitable for tissue engineering, as it can be molded into various shapes and has been used as a scaffold to support the growth and differentiation of diverse cell types [17]. Its biocompatibility and thermoresponsive properties make it an ideal choice for *in vivo* applications, undergoing gelation at physiological temperatures [18,19].

This study aimed to develop a modified protocol for fabricating a cell sheet composed of MDA-MB-231 cells and to investigate their ability to form tumors in an *in vivo* setting. By incorporating the cell sheet into Geltrex™, we provide a supportive scaffold that enhances mechanical support and promotes tissue integration. Importantly, the goal is not to alter the metastatic phenotype of MDA-MB-231 cells, which represent aggressive (TNBC), but rather to enable reproducible handling of these fragile cell sheets. Enhancing their structural integrity allows for consistent experimental manipulation, such as transplantation and *in vivo* modeling, without compromising the intrinsic biological characteristics of the cells. As an original research article, this work bridges methodological innovation with preclinical validation to address a critical technical barrier in TNBC modeling.

## 2. Materials and methods

### 2.1. Ethics of experimentation

The institutional research board at King Abdullah International Medical Research Centre (KAIMRC)/ King Saud Bin Abdulaziz University for Health Science (KSAU-HS) approved the study (#RC15/136R). The animal studies were performed following approval by The Institutional Animal Care and Use Committee (IACUC) at KAIMRC/KSAU-HS. All experimental protocols, including surgical procedures, anesthetics, analgesics, and other medications used, were approved by the ethical committee. Appropriately trained personnel performed all animal procedures in accordance with institutional and international ethical guidelines for the care and use of laboratory animals.

## 2.2. Cell lines culture

In this study, two different cell lines, MDA-MB-231 and Human Umbilical Vein Endothelial Cells (HUVECs), were utilized. BC cell line MDA-MB-231 (cell with poor cell-cell adhesions) was purchased from ATCC. The cells were cultured in Dulbecco's Modified Eagle Medium (DMEM) supplemented with 10% qualified fetal bovine serum NZ (FBS), 100 µg/ml of L-glutamate, 100 µg/ml streptomycin and 100 U/l penicillin (all from Invitrogen, Saudi Arabia) at $2 \times 10^5$ cells/cm2. Cells were maintained under cell culture conditions and incubated at 37°C in a humidified atmosphere of 5% $CO_2$. Then, harvested when 80% confluence was reached (approximately within 72 hours) by trypsinization and centrifugation at 500 xg for 10 minutes, followed by 200 x g for 5 minutes. The harvested cells were used in the subsequent procedure; construction of cancer cell sheets and cell suspension orthotropic injection.

HUVECs (Lonza, USA) cell line was selected as a control to assess and compare the integrity of the cell sheets. HUVECs were cultured in endothelial cell culture medium (EBM) (EGM, Lonza) supplemented with 2% FBS and vascular endothelial growth factor (VEGF) (Invitrogen). Cells were maintained under cell culture conditions and incubated at 37°C in a humidified atmosphere of 5% $CO_2$.

## 2.3. Standard protocol for fabrication of cell sheets

To construct the cell sheets, MDA-MB-231 cells (1.5 x 106) and HUVECs (1x106) were cultured separately in a 35 mm temperature-responsive polymer poly (N-isopropyl acrylamide)-coated plates (UpCell™, ThermoFisher Scientific, UK), and incubated at 37°C under 5% $CO_2$.

To determine the adhesion and proliferation patterns of the cell line cultivated on the UpCell™ culture dishes, the cells were seeded at the same density and incubated under the same conditions in tissue-culture polystyrene (TCPS) standard culture dishes of 35 mm in diameter.

Upon achieving confluence – (24-hour for HUVECs and 48-hour incubation for MDA-MB-231) – the media was removed, and the cells were gently washed once with 37°C Dulbecco's Phosphate-Buffered Saline (DPBS) (Invitrogen, Saudi Arabia) to eliminate dead cells. Therefore, cells were harvested as cohesive cell sheets through a low-temperature procedure at 20°C while maintaining a 5% $CO_2$ environment for 20 minutes. This method facilitated the cell sheet detachment from the culture dishes due to its temperature-responsive properties.

## 2.4. Geltrex™ modified protocol for fabrication of MDA-MB-231 cell sheet

Once the cell sheet was ready, on the day of transplantation, all the liquid in the UpCell™ culture dish was removed, and 250 µl of Geltrex™ was applied to the surface of the cell sheet (ThermoFisher Scientific, UK). Geltrex™ was used as an adhesive substrate to keep the cell sheet anchored during handling and transplantation. The Geltrex™ was left to dry for 1 hour at room temperature (RT). Next, a second layer of 500 µl of Gelatin (7.5% w/v, 0.75g in 10 mL dH2O, Al Alali, KSA) was added and left to sit for 10 minutes at RT, yielding a cell sheet construct. Gelatin was used as a carrier substrate to actually transfer the cell sheet construct.

To confirm cell viability, the cell sheet construct was re-cultured in a standard cell culture dish (TCPS) with the cell sheet facing down. An identical cultural environment was employed for conducting the cultural MDA-MB-231 experiment, as mentioned above. ImageJ (1.54d) was used for quantifying and analyzing the cell viability of the construct in comparsion to the standard MDA-MB-231 cell sheet.

To visualize cell morphology and structure, the cells were fixed with a mixture of methanol and acetic acid for 10 minutes at RT. After removing the fixing solution, crystal violet was used to stain the cells. In this study, we utilized crystal violet staining to assess overall cell density and adhesion within the fabricated cell sheets.

## 2.5. Animal

The animal studies were performed after receiving approval from the Institutional Animal Care and Use Committee (IACUC) in KAIMRC/KSAU-HS. The ethical committee approved all the experimental protocols for animals, including surgical procedures, anesthetics, and other medications used.

Adult female NU/J nude mice (8 weeks) were obtained from the Animal House Facility of King Saud University (Riyadh, Saudi Arabia). Mice were housed in polypropylene cages lined with husk under standard environmental conditions (temperature of 23–25°C, ambient humidity of 55–60%, and light: dark cycle of 12:12 hours). Mice were fed pellets and had unrestricted access to water. This study used three groups, each consisting of 3 mice. Group A (Negative control: n = 1): no treatment; Group B (Positive control): mice orthotopically injected with MDA-MB-231 cells ($5x10^6$) in the mammary fat pad; Group C (Test): mice transplanted subcutaneously with cell sheet constructed from MDA-MB-231 with carrier (Geltrex™ plus Gelatin). The sample size was calculated using an online sample size calculator by "Creative Research Systems" [20]. Mice were provided a standard pellet diet and had ad libitum access to water. Animals were regularly monitored by trained personnel for any signs of distress, pain, or abnormal behavior.

## 2.6. Surgical procedures and anesthesia

For both the transplantation of the Geltrex™-Gelatin modified MDA-MB-231 cell sheet and the orthotopic injection of MDA-MB-231 cell suspension, mice were anesthetized via subcutaneous injection of a Xylazine (10 mg/kg) and Ketamine (100 mg/kg) mix. Anesthesia depth was confirmed by the absence of reflex responses to tactile and painful stimuli before the procedure. Post-surgical recovery was closely monitored, with mice housed individually to prevent cannibalism or injury.

### 2.6.1. Transplantation of Geltrex™ Gelatin modified MDA-MB-231 cell sheet into nude mice.
Using the following procedure, the cancerous cell sheet construct was implanted subcutaneously into the flanks of nude mice. The mice were initially subjected to anesthesia by subcutaneously injecting 0.2 ml of Xylazin/Ketamine mix at a dose of 20 mg/kg and 100 mg/kg body weight, respectively. The dorsal skin was disinfected using Povidone-iodine ("betadine"). Subsequently, a longitudinal incision was made along the midline of the dorsal skin and detached from the surrounding tissue using straight surgical scissors. The transplant was positioned beneath the dermis layer using a supportive membrane, allowing for optimal adherence to the underlying subcutaneous tissue for 3 minutes. Finally, the incision was then sutured with 4−0 nylon thread.

## 2.7. Orthotopic injection of cancer MDA-MB-231 cell suspension

The mice were anesthetized by subcutaneously injecting 10 mg/kg and 100 mg/kg of Xylazin and Ketamine, respectively. A minor surgical procedure was performed by creating a small incision with scissors between the fourth nipple and the midline, and a cotton swab moistened with normal saline was inserted to form a cavity. The fat pad was observed based on its characteristic white coloration. The mammary fat pad was squeezed using a tweezer, resulting in complete exposure of the fat pad. Subsequently, 85 μl of the cell suspension ($5 \times 10^6$ cells) was injected into the mammary fat pad via a horizontal needle orientation.

## 2.8. Post-surgery care

Immediately after the surgery, the mice were housed individually to avoid cannibalism or suffocation. The mice were monitored regularly (at least every 15 minutes) until it is conscious and fully ambulant. Then, they were assessed daily for 14 days to ensure that there were no complications. To control moderate to severe post-surgery pain, if any, the animals were injected subcutaneously with buprenorphine (0.01–0.05 mg/kg) every 6–8 hours for a minimum of 48–72 hours. Mice were allowed to gradually increase their activity levels as tolerated, under the supervision of appropriately trained personnel. Palpation was used to monitor tumor development and progression.

## 2.9. Methods of sacrifice

At the end of the experiment, humane euthanasia was performed following AVMA (American Veterinary Medical Association) guidelines for the euthanasia of laboratory animals. Mice were euthanized using $CO_2$ inhalation followed by cervical dislocation to ensure a humane and painless sacrifice. Proper euthanasia was confirmed by the absence of respiration and heartbeat, followed by immediate tissue collection for histological and molecular analysis.

## 2.10. Humane endpoints

Animals were monitored for predefined humane endpoints, including excessive weight loss (>20% body weight reduction), severe ulceration or necrosis at the tumor site, impaired mobility, lethargy, and signs of distress. If an animal exhibited severe clinical symptoms, euthanasia was performed immediately using $CO_2$ inhalation to prevent unnecessary suffering.

This study adheres to internationally recognized ethical principles for the care and use of animals in research, ensuring that all efforts were made to minimize pain, distress, and suffering while maximizing the scientific value of the study.

## 2.11. Histological staining

**2.11.1. Hematoxylin and eosin (H&E) staining.** Tissue sections (4μm) were prepared for hematoxylin and eosin (H&E) staining using SAKURA Prisma slide stainer. The sections were deparaffinized using xylene, followed by rehydration with a series of graded alcohols. Hematoxylin staining was performed to visualize the nuclei within the tissue, followed by eosin to highlight the cytoplasmic components. Therefore, dehydration of the stained sections was accomplished by passing them through a series of graded alcohol, and finally cleared with xylene and mounted. A pathologist examined the sections under a light microscope to evaluate tissue morphology, cellular details, and pathological changes.

## 2.12. Statistical analysis

A two-sample t-test was conducted to compare the viability of cells in two different groups. The statistical analysis was performed using GraphPad Prism (version 5.03). A two-sided *P*-value of <0.05 was considered statistically significant.

## 3. Results

### 3.1. Standard MDA-MB-231 cell sheet fabrication

Under optimal culture conditions on standard 35mm tissue culture polystyrene (TCPS), MDA-MB-231 cells; representing a (TNBC) model, doubled in number within 48 hours.

Upon reaching confluence (approximately 85–90%), the cells successfully formed an intact sheet using the UPcell™ temperature-responsive dish, Fig 1a. However, despite confluence, these native MDA-MB-231 cell sheets exhibited considerable fragility upon detachment. This fragility manifested in poor mechanical stability, limited cohesion, and frequent tearing upon handling or transplantation, Fig 1d. This behavior contrasts with cell sheets fabricated from human umbilical vein endothelial cells (HUVECs), which formed structurally robust, highly cohesive sheets, Fig 1e. The reduced mechanical integrity and junctional cohesion observed in MDA-MB-231 sheets may be attributed to their mesenchymal-like phenotype and inherently weak cell-cell junctions—hallmarks of epithelial-to-mesenchymal transition (EMT) common in TNBC subtypes.

To overcome these structural limitations, we developed a modified fabrication protocol incorporating Geltrex™ and Gelatin to generate a reinforced cell sheet construct. This adaptation significantly improved sheet integrity, enhanced intercellular adhesion, and promoted the formation of a tissue-like structure as in Fig 2. Compared with native MDA-MB-231 sheets, the construct showed better resistance to tearing, increased mechanical stability, and ease of transplantation, while retaining the biological properties of the parent cell line.

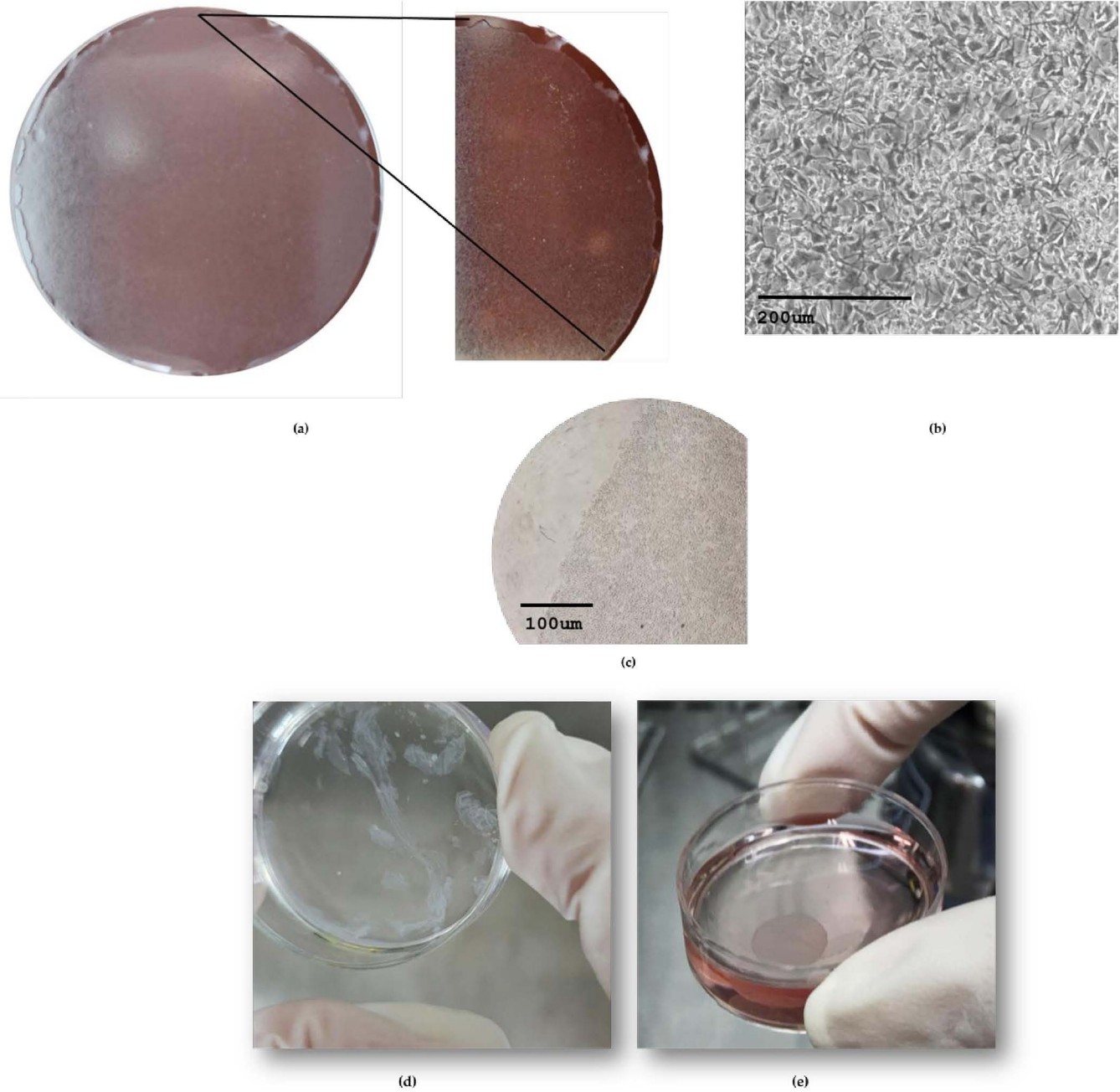

**Fig 1. Representative images of cell sheets in Upcell dish.** (a) photograph of the intact MDA-MB-231 cell sheet cultured before detachment; (b) Morphologies of MDA-MB-231 cell sheet at 20x magnification before detachment; (c) Microscopic image of the edge of the MDA-MB-231 cell sheet, revealing the starting of detachment from the Upcell dish at 10x magnification; (d) Broken and fragile sheet fabricated from MDA-MB-231 after detachment; (e) Well-performed and intact cell sheet fabricated from Human Vascular Endithelial Cells (HUVECs) after detachment.

Cell viability analysis using ImageJ (v1.54d) on stained images revealed that the cell sheet construct supported a higher number of viable cells (349) compared with the unmodified MDA-MB-231 sheet (280), though the difference was not statistically significant (P > 0.05). Notably, cells from the construct-maintained viability and proliferative capacity during

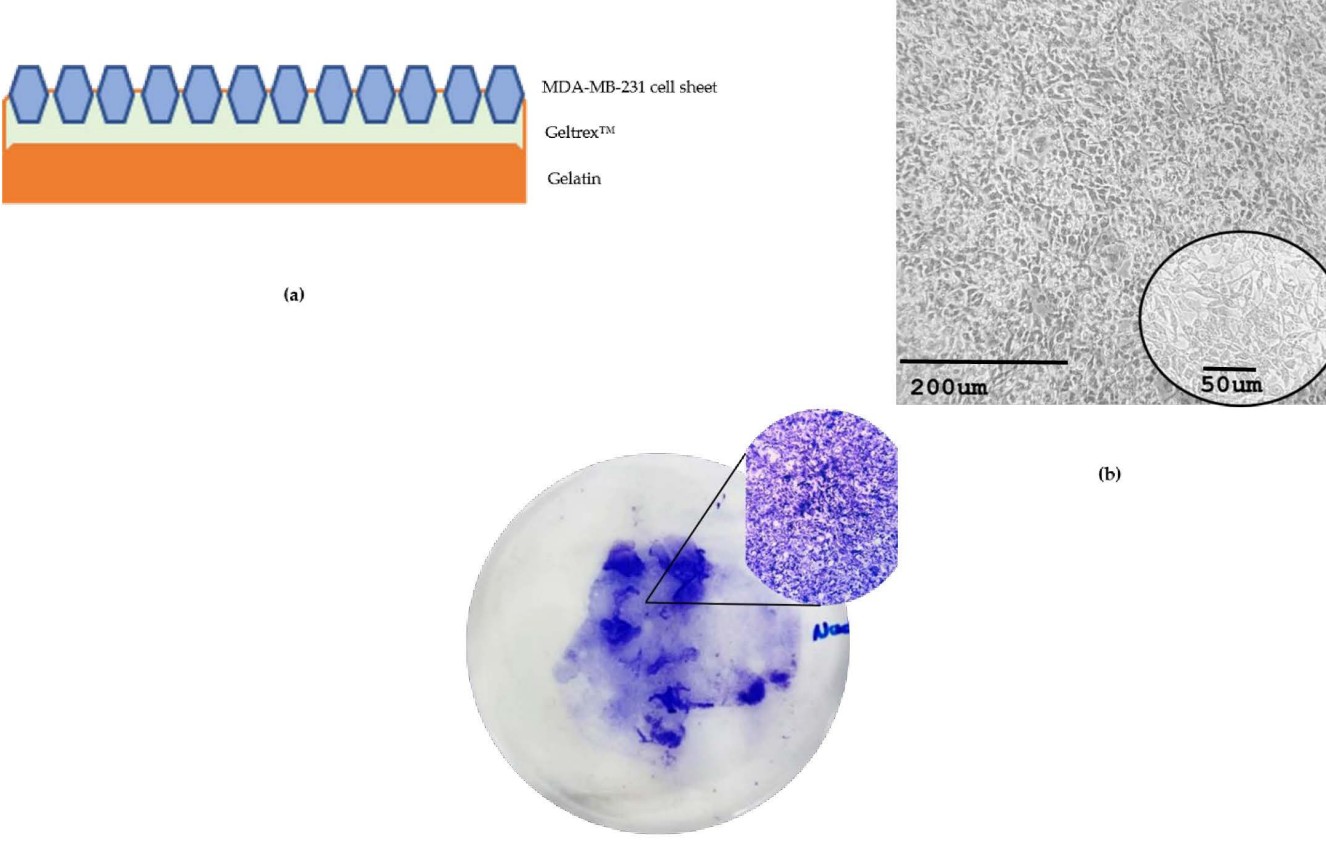

**Fig 2. Modified method of MDA-MB-231 cell sheet using Geltrex™ and Gelatin.** (a) Diagram illustrating the layer of the construct of MDA-MB-231 cell sheet+ Geltrex™ + Gelatin; (b) Morphologies of the construct at 20x and 40x magnifications; (c) Morphology of the MDA-MB-231 cell sheet+ Geltrex™ construct is analyzed by Crystal violet staining.

re-culturing, indicating structural and functional preservation post-detachment shown in Fig 3. Crystal violet staining further confirmed the retention of cancer cell morphology within the construct, supporting its suitability for downstream applications. It is important to note that crystal violet primarily stains DNA and does not distinguish between live and dead cells. While this assay serves as a qualitative indicator of cell presence and general sheet integrity, we recognize the limitations in using it as a direct measure of cell viability. Future studies could benefit from incorporating additional viability assays.

Given the aggressive nature of TNBC and the lack of well-defined molecular biomarkers or targeted therapies, models that accurately recapitulate the tumor microenvironment are critical. The enhanced structural fidelity of this cell sheet construct presents a promising platform for testing therapeutic responses, particularly in strategies targeting the extracellular matrix, cell adhesion molecules, or signaling pathways involved in EMT and metastasis. Furthermore, this model may be adapted to study drug resistance and biomarker discovery in TNBC, offering a physiologically relevant scaffold that bridges the gap between conventional 2D cultures and complex *in vivo* systems. While this study qualitatively demonstrated improved structural integrity of modified cell sheets, such as reduced tearing and enhanced cohesion, we acknowledge the absence of quantitative metrics, such as tensile strength or rheological measurements. to clarify that the MDA-MB-231 cell line used in this study known to have weak or absent cell junctions [12], which poses a significant

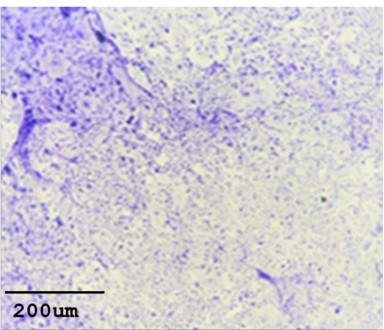

**Fig 3. The MDA-MB-231 cell sheet+ Geltrex™ construct after 24 hours culture in a standard (TCPS) cell culture dish with the cell sheet facing down in DMEM medium and stained by Crystal violet.**

challenge in assessing structural integrity using conventional quantitative methods in our model. However, qualitative observations align with functional outcomes, including successful engraftment and tumor formation (Fig 4).

### 3.2. Tumourigenesity of the transplanted modified MDA-MB231 cell sheet (cell sheet construct) vs. orthotopic injection of MDA-MB-231 cell suspension

Using Gelatin to coat the Geltrex™ -coated MDA-MB-231 cell sheet (cell sheet construct) provided stability for easy manipulation using forceps, allowing precise placement in the target area. This resulted in successful engraftment and integration of the cells into the animal tissue.

Within eight weeks, the average tumor size resulting from the subcutaneous transplantation of the cell sheet construct was measured to be approximately 5.7 mm. Conversely, the average tumor size in the mammary fat pad following cell injection was slightly larger at 5.8 mm during the exact timeframe (Fig 4).

### 3.3. Histological evaluation of collected tissues

Both methods successfully induced BC in mice. For the orthotopic injection of MDA-MB-231 cell suspension method, H&E stained fatty pad tissue sections demonstrated poorly differentiated malignant cells/carcinoma with extensive geographic necrosis in more than 60% of the tissue. The total score corresponding with grade 3 is based on the Nottingham/ modified Bloom & Richardson Score (Fig 5a).

While, in the cell sheet construct transplant method, stained cells demonstrated poorly differentiated malignant cells/ carcinoma with mild geographic necrosis in less than 10% of tissue—the tumor cells infiltrative fat and skeletal muscle. The total score corresponding with grade 3 is based on the Nottingham/ modified Bloom & Richardson Score (Fig 5b).

Another notable difference that emerged between the two techniques was metastasis. The modified cell sheet method displayed a higher incidence of metastasis, particularly in the liver (Fig 6a) and spleen (Fig 6b), compared to the injection method.

## 4. Discussion

The cell sheet technique has shown promising results in developing preclinical models for BC, as it allows for the generation of three-dimensional tissue constructs that better mimic the complexity of human BC more accurately to reflect tumor architecture and cellular behavior. However, this technique's success relies on the cell sheet's ability to maintain its integrity during transplantation. This is especially challenging when using cell lines like MDA-MB-231, which are characteristic of (TNBC) and known for their poor cell-cell junctions due to mesenchymal-like properties driven by

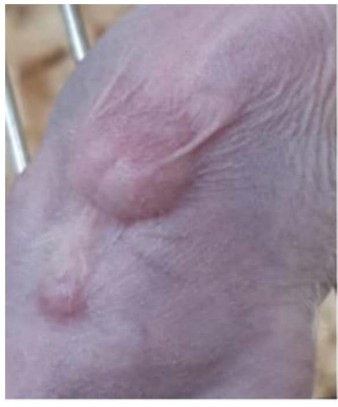 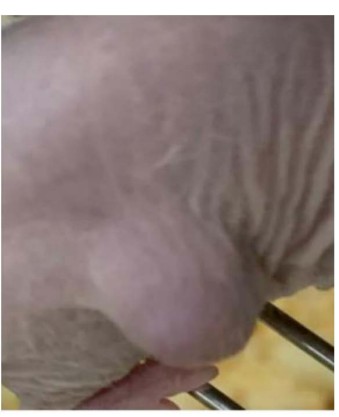

(a)                                    (b)

**Fig 4. Comparison of tumor growth in mice using two different methods after eight weeks.** (a) Subcutaneous cell sheet construct transplantation resulted in the formation of a tumor average size of 5.7 mm; (b) Orthotopic injection of MDA-MB-231 cell line into the mammary fat pad led to the development of a tumor average size of 5.8 mm.

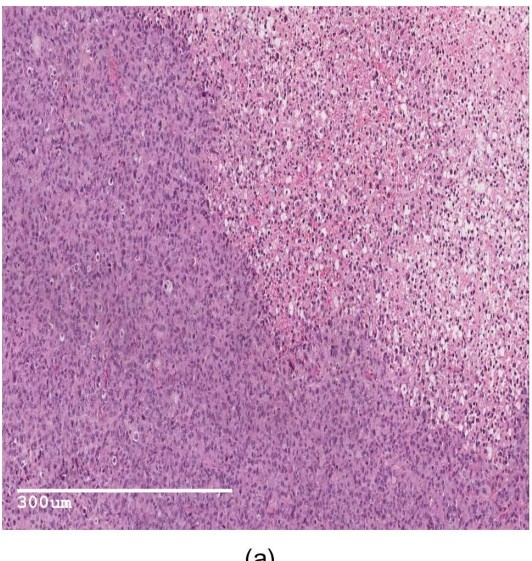 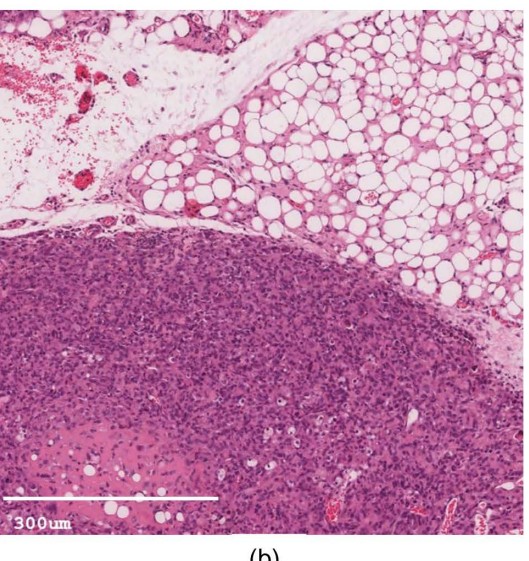

(a)                                    (b)

**Fig 5. H&E staining slide illustrating the developed tumor in mice after 8 weeks.** (a) using orthotopic mammary fat pad MDA-MB-231 cell injection; (b) using cell sheet construct subcutaneous transplant.

epithelial-to-mesenchymal transition (EMT) mechanisms [12]. EMT leads to the downregulation of junctional proteins such as E-cadherin, compromising cell cohesion and making these cells poorly suited for forming stable sheets. These structural limitations reduce handling reproducibility, hinder *in vivo* implantation, and can confound experimental outcomes.

In this study, we aimed to address this issue by incorporating Geltrex™ plus Gelatin (construct) during cell sheet fabrication protocol. Geltrex™, a basement membrane extracts rich in ECM proteins (e.g., laminin, collagen IV, and entactin), serves to mimic the native tumor microenvironment, supporting cell adhesion, survival, and matrix remodeling.

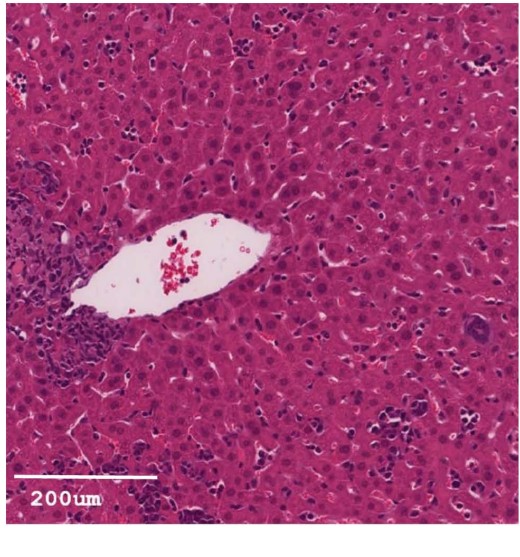

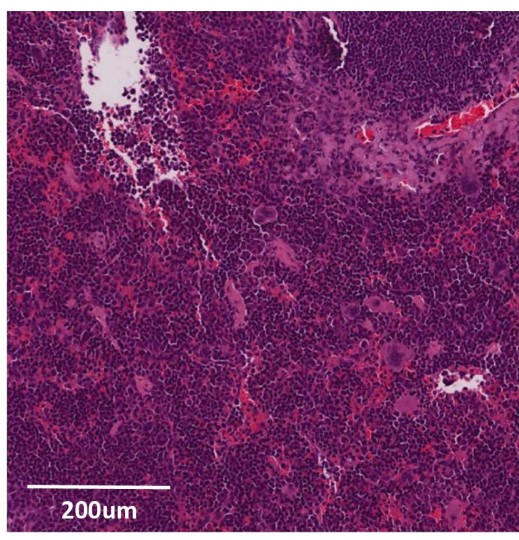

(a)
(b)

**Fig 6. H&E staining slide illustrating the metastasis in mice using cell sheet construct subcutaneous transplant after 8 weeks.** (a) Liver; (b) Spleen.

While Geltrex™ has not explicitly been used with cell sheets to develop cancer in animal models, it has been used in several studies to create 3D culture systems to study cancer and develop anticancer therapies [21,22]. Geltrex™ mimics the extracellular matrix and promotes cell adhesion [23]. It provides biological cues through integrin-mediated signaling pathways, which may enhance focal adhesion formation and indirectly stabilize intercellular junctions. Gelatin, on the other hand, is a more transient support that acts as a temporary adhesive that facilitates the attachment of the cell sheet construct to the animal model, which readily dissolves once *in vivo* [24]. This construct encapsulated the cell sheet prior to transfer, providing a scaffold that supports the cell sheet and prevents fragmentation. Therefore, Geltrex™ and Gelatin both serve an optimal function at different process stages. Together, this construct improved mechanical integrity and sheet cohesion without altering the inherent metastatic properties of MDA-MB-231 cells; aligning with our aim to support, not reverse, the phenotype.

The use of Geltrex™ in cell sheet fabrication offers significant advantages in promoting cell adhesion and enhancing sheet integrity. However, it is important to acknowledge the potential cost implications associated with its use. Geltrex™ is a proprietary product, and its pricing can vary significantly based on batch production and supplier. This variability can lead to challenges in maintaining consistent experimental conditions and may impact the scalability of protocols in larger studies or clinical applications. Scalable alternatives such as synthetic polymers (e.g., polyethylene glycol [PEG]-based hydrogels) or decellularized matrices could mitigate cost and variability concerns [7,17]. Future optimization will explore these materials to enhance accessibility without compromising functionality.

Furthermore, the reliance on a commercially available matrix like Geltrex™ can increase overall experimental costs, particularly for laboratories with limited funding or those aiming for high-throughput applications. It is crucial for researchers to consider these factors when planning studies that may require repeated use of Geltrex™ or similar matrices.

Data from this study revealed that the modified protocol improved the transfer efficiency of cell sheets derived from a low junction MDA-MB-231 cell line into a mouse BC model. This resulted in superior engraftment efficiency of the cell sheet and the formation of tumors that closely resembled human BC. First, despite the differences in the number of cells seeded between the two methods (5 × 10$^6$ cells vs. 1.5 x 10$^6$ cells; duplication time is 48 hrs) the average

tumor size resulting from the cell sheet transplantation was comparable to that observed with the cell injection method. This suggests that the modified cell sheet technique is as effective as the conventional method in inducing tumor growth within the experimental timeframe. One possible explanation for the similar tumor sizes could be the ability of the transplanted cell sheets to integrate with the surrounding tissue, enabling the cancer cells to proliferate and form tumors. This integration may compensate for the subcutaneous location of the cell sheet transplantation, which differs from the direct injection into the mammary fat pad as performed in the cell injection method. Second, one of the disadvantages associated with the mammary fat pad injection technique is it introduces cells directly into the fatty pad, bypassing the initial formation of a cohesive tumor structure. This technique may result in a more disorganized tumor microenvironment, limiting the ability of tumor cells to establish metastatic colonies in distant organs. On the other hand, the potential loss of injected cells into the circulation; these cells can travel to distant organs and create the appearance of metastatic spread, even though they may not possess the inherent capacity for metastasis. While the cell sheet method involves implanting a cohesive cell layer, which may better mimic the complex three-dimensional architecture of breast tumors. This structural similarity could potentially facilitate the migration and dissemination of tumor cells, leading to an increased likelihood of metastasis. In the current study, the observed metastasis to the liver in the cell sheet method aligns with clinical observations of BC patients, where liver metastases are the most common sites of distant metastasis [25].

This platform holds specific promise for modelling TNBC, a subtype lacking targeted therapies. The scaffold-supported cell sheets offer a physiologically relevant platform to investigate TNBC-specific molecular markers (e.g., EGFR, PD-L1, c-Met) and to evaluate the efficacy of emerging targeted therapies such as EGFR inhibitors, PARP inhibitors, and immune checkpoint modulators [26,27]. Additionally, TNBC often exhibits basal-like or mesenchymal phenotypes, driven by genetic and epigenetic alterations in key pathways such as PI3K/AKT, Wnt/β-catenin, Notch, and EGFR [28,29]. Therefore, the scaffold-supported sheets could elucidate TNBC-specific pathways such as Wnt/β-catenin and Notch, as well as mechanisms of drug resistance, bridging gaps between 2D models and clinical complexity.

From a mechanistic standpoint, providing physical support through Geltrex™ may also promote focal adhesion kinase (FAK) signaling, known to regulate actin cytoskeleton dynamics and metastatic signaling in TNBC. While not directly studied here, this opens new avenues for future research into how scaffold stiffness and composition modulate signaling pathways involved in migration, invasion, and chemoresistance. The model could also be adapted to include co-culture systems with stromal or immune cells to further recapitulate the tumor microenvironment, making it suitable for high-content drug screening or immune-oncology applications.

The dual approach of cell sheet and cell injection facilitated a comprehensive analysis of BC progression, providing valuable insights for further understanding the disease and developing effective therapies. Although the present study did not explore the behavior of cell sheets in the mammary fat pad model, it serves as a stepping-stone for future research. Before delving into the mammary fat pad model, it was crucial to verify the efficacy of the modified protocol and assess the viability and morphology of the cancer cells in the fabricated cell sheets. Further investigation is warranted to study the behavior of cell sheets in the mammary fat pad model and evaluate their potential for application in BC research. The findings suggest that Geltrex™ plus Gelatin could be a valuable tool for improving the transfer efficiency of the cell sheets in the development of preclinical models for BC. Further studies are needed to optimize these biomaterials' use and determine their potential application in other types of cancer. As effective models are crucial in BC research to replicate disease progression and treatment outcomes accurately.

Beyond BC modeling, this platform has broader implications in tissue engineering and personalized oncology. Cell sheets could be generated using patient-derived cells, enabling the creation of individualized tumor models for biomarker discovery, drug sensitivity testing, and mechanistic studies. The findings from this study establish a foundation for using scaffold-supported sheets to improve reproducibility and physiological relevance in preclinical cancer research, without compromising the native behavior of aggressive cancer phenotypes like TNBC.

## 5. Conclusions

The modification of the protocol using Geltrex™ and Gelatin has effectively addressed a key technical barrier in the use of cell sheets derived from breast cancer (BC) cells with poor intercellular junctions, such as the MDA-MB-231 line. By enhancing the structural integrity and transfer efficiency of these fragile sheets, the approach improves reproducibility and handling, making it more viable for translational applications. Importantly, this strategy preserves the aggressive and metastatic phenotype characteristic of (TNBC), allowing for more physiologically relevant *in vivo* modeling without compromising biological fidelity.

Beyond technical refinement, this work offers broader biological and translational implications. The enhanced cell sheet integrity supports the development of more physiologically relevant *in vitro* and *in vivo* models that better replicate the tumor microenvironment. Such models could serve as valuable tools for preclinical drug screening, therapeutic response assessment, and mechanistic studies, especially for aggressive cancers like TNBC that currently lack targeted therapies.

Looking forward, this approach could be adapted for use with patient-derived cancer cells, paving the way for personalized cancer modeling and individualized drug testing. Additionally, the success of this scaffold-enhanced technique holds promise in regenerative medicine, with potential applications in the engineering of tissues such as skin, cartilage, and bone. By improving the functionality and reliability of cell sheet constructs, this study lays the groundwork for future innovations in cancer research and tissue engineering, offering new avenues to improve patient outcomes.

Overall, this work establishes a platform that not only enhances experimental modeling of aggressive cancers but also supports future innovations in scaffold-assisted tissue reconstruction. By bridging a critical gap between cell culture and *in vivo* modeling, the modified cell sheet protocol offers a valuable tool for both oncological research and clinical translation.

## Acknowledgments

We would like to thank the administrative staff at King Abdullah International Medical Research Center (KAIMRC) and King Saud University, Riyadh, Saudi Arabia.

## Author contributions

**Conceptualization:** Alaa T. Alshareeda, Nada Albarakati.

**Data curation:** Alaa T. Alshareeda, Nada Albarakati, Ayidah Alghuwainem, Sarah Al-Maiman, Amal S. Alhamid.

**Formal analysis:** Alaa T. Alshareeda, Nada Albarakati, Amal S. Alhamid.

**Funding acquisition:** Alaa T. Alshareeda.

**Investigation:** Alaa T. Alshareeda, Nada Albarakati, Yasser Alshawakir, Ayidah Alghuwainem, Abdul Latif Khan, Abdullah Almubarak.

**Methodology:** Alaa T. Alshareeda, Nada Albarakati, Yasser Alshawakir, Ayidah Alghuwainem, Batla S. Al-Sowayan, Abdul Latif Khan, Abdullah Almubarak, Ahood Al Sayed.

**Project administration:** Alaa T. Alshareeda.

**Resources:** Alaa T. Alshareeda.

**Supervision:** Alaa T. Alshareeda.

**Validation:** Abdul Latif Khan, Sarah Al-Maiman, Amal S. Alhamid.

**Visualization:** Alaa T. Alshareeda.

**Writing – original draft:** Alaa T. Alshareeda, Nada Albarakati, Batla S. Al-Sowayan, Nur Khatijah Mohd Zin.

**Writing – review & editing:** Alaa T. Alshareeda, Nada Albarakati, Yasser Alshawakir, Ayidah Alghuwainem, Batla S. Al-Sowayan, Abdul Latif Khan, Abdullah Almubarak, Sarah Al-Maiman, Ahood Al Sayed, Nur Khatijah Mohd Zin, Amal S. Alhamid.

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
