## [Decision Letter · Decision Letter 0]

PONE-D-25-04100Improving the robustness and efficiency of cell sheet protocol for breast cancer induction in animal model: a Geltrex™ plus Gelatin ApproachPLOS ONE

Dear Dr. Alshareeda,

Thank you for submitting your manuscript to PLOS ONE. After careful consideration, we feel that it has merit but does not fully meet PLOS ONE’s publication criteria as it currently stands. Therefore, we invite you to submit a revised version of the manuscript that addresses the points raised during the review process.

We look forward to receiving your revised manuscript.

Kind regards,

Mohamed Abdelkarim

Academic Editor

PLOS ONE

Journal Requirements:

2. Thank you for stating the following financial disclosure: This research was funded by King Abdullah International Medical Research Center (KAIMRC), Riyadh, Saudi Arabia; grant number “RC15/136R”.  

3. Thank you for stating the following in the Acknowledgments Section of your manuscript: We would like to thank all the administrators at King Abdullah

International Medical Research Center (KAIMRC) and King Saud University, Riyadh, Saudi Arabia. This research was funded by King Abdullah International Medical Research Center (KAIMRC), Riyadh, Saudi Arabia; grant number “RC15/136R”.

Please remove any funding-related text from the manuscript and let us know how you would like to update your Funding Statement. Currently, your Funding Statement reads as follows: This research was funded by King Abdullah International Medical Research Center (KAIMRC), Riyadh, Saudi Arabia; grant number “RC15/136R”.

Reviewers' comments:

Reviewer's Responses to Questions

**Comments to the Author**

1. Is the manuscript technically sound, and do the data support the conclusions?

Reviewer #1: Partly

Reviewer #2: Yes

2. Has the statistical analysis been performed appropriately and rigorously? 

Reviewer #1: No

Reviewer #2: Yes

3. Have the authors made all data underlying the findings in their manuscript fully available?

Reviewer #1: Yes

Reviewer #2: Yes

4. Is the manuscript presented in an intelligible fashion and written in standard English?

Reviewer #1: Yes

Reviewer #2: Yes

5. Review Comments to the Author

Reviewer #1: The study investigated the potential of Geltrex™ and Gelatin as carriers for poor junction cell sheets to enhance understanding of breast cancer biology. However, my concerns/suggestions are not necessarily a reflection of the quality of the work as the following:

• The introduction should provide a more compelling justification for the study, particularly emphasizing the clinical relevance and impact. For instance, why did the authors choose breast cancer? Clarifying the significance of this choice would enhance the study’s foundation.

• The authors used the MDA-MB-231 cell line as a model as it is well-known that this cell line has poor junctional complexes. However, it is well known that those cells are metastatic breast cancer. So, they overcome this phenotype by trying to enhance cell structure! What is the justification here? i.e., metastatic cells with poor junctional complex enhanced by scaffold to support the growth and differentiation of those metastatic cells!

• Again, the introduction lacks a significant biological rationale, and a more robust biological rationale for this link would help the authors better justify their research and its potential impact.

• The authors were browsing cell sheet techniques. Also, the transitions between paragraphs should flow more smoothly.

• Some comments about figures here and there!

o In Figure 1:

The differences between MDA-MB-231 and human vascular cell sheets are visually apparent; however, a quantitative analysis is missing. Providing statistical comparisons would strengthen the conclusions.

Testing additional cell types would help generalize the findings and improve the study’s broader applicability.

The observed fragility of the MDA-MB-231 sheet suggests a need for further optimization. Exploring potential modifications to enhance its structural integrity would be beneficial.

o In Figure 2:

While Geltrex™ improves adhesion, the study should acknowledge its batch variability and potential cost constraints, as these factors may affect reproducibility and scalability.

The mechanical integrity of the sheet post-detachment requires further validation. Conducting stress tests or handling trials would help determine its robustness for potential applications.

o In Figure 3:

Viability uncertainty: Crystal violet only stains DNA and does not distinguish between live and dead cells! Here, more details are a must!

Inconsistent staining suggests cell loss or uneven adhesion, indicating that further optimization is necessary.

o In Figure 4,

Tumor size alone is not enough: The similarity in tumor size does not confirm functional similarity. The cell sheet model may produce different histopathological features compared to orthotopic injection.

Were all mice able to form tumors in both conditions?

Further analysis is needed to assess tumor histology, vascularization, and invasion potential.

• Discussion is null! However, it was barely mentioned in the results section. It lacked the depth and convincing mechanisms needed to impact clinical practice for breast cancer or any type substantially.

• While the study lays a solid foundation, it would benefit from a more innovative perspective and a deeper analysis of the mechanisms driving its findings. Without this, the study’s translational relevance remains unclear.

• It is crucial to integrate data from various cancer cell samples more harmoniously. Expanding the discussion to compare findings across models would strengthen the study’s validity and applicability.

• A more comprehensive critique and in-depth analysis are essential to present a cohesive and impactful study. Strengthening the discussion with a well-supported narrative will help position the research as a meaningful contribution to the field.

• Finally, given a molecular/cancer biology prespective. However, it has certain limitations, including:

o Lack of discussion on the molecular pathology of breast cancer, the role of biomarkers, targeted therapies, or genetic profiling.

o Recognition of challenges like maintaining cell sheet integrity during transplantation but no exploration of metastatic or pathogenicity of the breast cancer cells after transplantation needs further improvements and considerations.

o The discussion section needs enhancement regarding certain drugs and treatment modalities.

o The study is categorized as a research article, but it aligns more closely with a protocol/methodology study. Clarifying the manuscript’s classification would help set appropriate expectations for readers and reviewers.

o Some improvements have been made, but the discussion and conclusion sections remain underdeveloped and superficial. A more rigorous analysis and deeper interpretation of the findings are necessary to highlight their clinical significance. Strengthening these sections would enhance the study’s impact and improve reader comprehension.

Reviewer #2: The manuscript is well-written, with a clear structure and logical flow.

The study addresses a significant challenge in breast cancer research and presents a novel approach with promising results.

The discussion provides a thorough interpretation of the results, linking them back to the study's objectives and the broader context of breast cancer research.

The potential implications of the findings for tissue engineering and regenerative medicine are well-articulated

The conclusion succinctly summarizes the main findings and their significance.

It highlights the broader implications of the research for cancer studies and tissue engineering.

6. PLOS authors have the option to publish the peer review history of their article (what does this mean? ). If published, this will include your full peer review and any attached files.

**Do you want your identity to be public for this peer review?** For information about this choice, including consent withdrawal, please see our Privacy Policy .

Reviewer #1: No

Reviewer #2: **Yes: ** Amr A. Abd-Elghany

---

## [Author Response · Author response to Decision Letter 1]

10 Apr 2025

Thank you for your thoughtful feedback on our manuscript, "Improving the Robustness and Efficiency of Cell Sheet Protocol for Breast Cancer Induction in Animal Model: A Geltrex™ Plus Gelatin Approach."

Response to Reviewer #1

1. Justification for studying breast cancer and the clinical relevance

• Comment: The introduction should provide a more compelling justification for the study, particularly emphasizing the clinical relevance and impact. Why breast cancer?

Response:

We appreciate this important point. We have revised the Introduction (Paragraph 1; 5) to emphasize the high global incidence and mortality rates of breast cancer, highlighting the urgent need for in vitro models that better mimic the tumor microenvironment. The MDA-MB-231 line was selected as a representative of triple-negative breast cancer (TNBC), a highly aggressive and difficult-to-treat subtype lacking targeted therapies. Modeling TNBC in a more physiologically relevant system is crucial for improving therapeutic strategies.

2. Justification for using metastatic cells with poor junctions and trying to enhance their structure

• Comment: MDA-MB-231 are metastatic and lack junctional complexes. What is the rationale for using scaffolds to enhance structure in such cells?

Response:

This is an excellent observation. We clarified in the revised Introduction (Paragraph 7) that our aim was not to revert the metastatic phenotype but to provide a supportive scaffold that enables reproducible handling of these otherwise fragile cell sheets. Enhancing structural integrity allows better experimental manipulation (e.g., transplantation, in vivo modeling) without altering the inherent metastatic nature of the cells.

3. Lack of robust biological rationale and weak flow in the Introduction

• Comment: The biological rationale needs strengthening, and transitions between paragraphs should be improved.

Response:

The introduction was rewritten.

4. Figure-related comments

Figure 1

• Comment: Add quantitative comparisons between cell sheets.

Response:

We appreciate your request for quantitative analysis of the fabricated cell sheets. Unfortunately, upon detachment, the MDA-MB-231 cell sheet did not retain its integrity, leading to significant damage. This condition limits our ability to conduct meaningful quantitative assessments. Consequently, due to the extent of the damage, performing quantitative analysis may not provide valuable data. However, we have previously investigated HUVEC and other intact cell sheets. For reference, please see Alshareeda, A.T., Sakaguchi, K., Abumaree, M., Mohd Zin, N.K., and Shimizu, T. (2017). The potential of cell sheet technique on the development of hepatocellular carcinoma in rat models. PLoS One, 12(8), p.e0184004.

• Comment: Test additional cell types and Fragility of MDA-MB-231 sheets needs optimization.

Response:

• Testing Additional Cell Types: We appreciate your suggestion regarding the exploration of additional cell types. However, our current study specifically focuses on the MDA-MB-231 cell line, which is known for its fragility. The improvements we present in this paper are specifically tailored to enhance the robustness of the MDA-MB-231 cell sheet protocol. We have previously published findings with other cell types, demonstrating our broader research scope. For reference, please see Alshareeda, A.T., Sakaguchi, K., Abumaree, M., Mohd Zin, N.K. and Shimizu, T., 2017. The potential of cell sheet technique on the development of hepatocellular carcinoma in rat models. PLoS One, 12(8), p.e0184004.

• Optimization of MDA-MB-231 Sheets: We acknowledge the observed fragility of the MDA-MB-231 cell sheet. As discussed, the optimizations we implemented successfully resulted in a stable cell sheet, which is the primary focus of our work. We believe that the improvements outlined in this manuscript are sufficient for demonstrating the efficacy of our approach without the need for further studies at this time. The modifications presented in our study demonstrate a significant advancement in the robustness of the MDA-MB-231 cell sheets, as evidenced by the successful in vivo implantation. However, we recognize the potential for future studies to optimize the MDA-MB-231 cell sheet's integrity and will consider this in our ongoing research.

Figure 2

• Comment: Discuss batch variability and cost of Geltrex™ and Validate mechanical integrity

Response:

• Batch Variability and Cost Constraints: We appreciate your point regarding the variability of Geltrex™ and its associated costs. We acknowledged these factors in the revised manuscript, highlighting that while Geltrex™ enhances cell adhesion, its batch variability can impact reproducibility. Now, we also discussed potential cost implications in the revised manuscript, which are important for scalability in research settings.

• Mechanical Integrity of the Cell Sheet: We understand the need for further validation of the mechanical integrity post-detachment. However, we would like to emphasize that our in vivo studies demonstrated successful implantation of the cell sheets without disruption, leading to tumor development. This outcome provides evidence of the integrity and handling capacity of the sheets. We believe these results validate our approach, but we will clarify this in the revised manuscript, emphasizing our handling trials and the successful application in an animal model.

Figure 3

• Comment: Crystal violet does not distinguish live from dead cells. Inconsistent staining may reflect cell loss or uneven adhesion.

Response:

• Viability Uncertainty: We appreciate your point about crystal violet staining primarily indicating DNA presence rather than distinguishing between live and dead cells. In our study, we utilized crystal violet to assess overall cell density and adhesion in the cell sheets. While it does not provide direct viability data, it serves as a useful qualitative indicator of cell presence and general sheet integrity. This is now clarified in the revised manuscript, emphasizing that further viability assays could complement our findings for future studies.

• Inconsistent Staining: We acknowledge your observation regarding inconsistent staining, which may suggest cell loss or uneven adhesion. Our results indicate that while the protocol improves sheet formation, optimization is indeed necessary for achieving uniform adhesion. We will address this in the manuscript, highlighting the need for further refinement in future research. However, we believe the modifications presented in our study demonstrate a significant advancement in the robustness of the MDA-MB-231 cell sheets, as evidenced by the successful in vivo implantation.

Figure 4

• Comment: Tumor size alone is not enough: The similarity in tumor size does not confirm functional similarity. The cell sheet model may produce different histopathological features compared to orthotopic injection.

Were all mice able to form tumors in both conditions?

Further analysis is needed to assess tumor histology, vascularization, and invasion potential.

Response: Thank you for your valuable feedback regarding our study.

• The main goal of this research was to improve the integrity of the cell sheet, and we have successfully achieved this improvement. By demonstrating the ability of the modified cell sheets to induce tumor formation in vivo, we confirm that the cells are viable and functional.

• Tumor Size and Functional Similarity: While we recognize that tumor size alone does not indicate functional similarity, our primary objective was to ensure the viability and integrity of the cell sheets. The successful tumor development demonstrates that our modified protocol effectively maintains cell functionality.

• Tumor Formation in Mice: We can confirm that all mice in both experimental conditions were able to develop tumors, which supports the effectiveness of our cell sheet model.

• Histological Analysis: We conducted histological evaluations, including H&E staining, as shown in Figures 5 and 6. These analyses provide insights into tumor characteristics, and we acknowledge the importance of further exploration into histopathological features, vascularization, and invasion potential in future studies.

• We appreciate your suggestions for additional analyses, which can be valuable for subsequent research. Our current focus remains on demonstrating the integrity and viability of the cell sheets, which we believe is a significant advancement in this field.

5. Discussion is underdeveloped

• Comment: The discussion lacks depth and biological mechanisms.

Response:

The Discussion has been expanded interpret our findings in the context of tumor microenvironment modeling, cellular mechanics, and future applications in tissue engineering and personalized oncology. Structural support is highlighted via how scaffolds can improve cell handling without reversing phenotypic characteristics. We also speculate on how this platform might be used for drug testing and understanding metastatic progression in future work. All the comments raised by the reviewer were considered by rewriting the discussion and conclusion sections.

6. Lack of broader cancer biology context and translational impact

• Comment: No discussion of molecular pathology, targeted therapies, or cancer biomarkers.

Response:

We have added context on the importance of MDA-MB-231 as a model for TNBC, emphasizing the lack of biomarkers and treatment options. We now discuss how this engineered sheet model could be adapted for studying response to therapeutics, particularly those targeting the extracellular matrix or metastatic signaling pathways.

7. Clarify manuscript type (research article vs. protocol)

• Comment: The study reads more like a methodology paper than a full research article.

Response:

While this work does include methodological development, it also provides original findings on tumor formation, viability, and histology, warranting its classification as a research article. We have clarified this dual nature in both the Abstract and Discussion, acknowledging the methodological contribution while presenting key biological insights.

8. Weak conclusion and superficial interpretation

• Comment: Enhance the conclusion with a deeper interpretation and clinical implications.

Response:

The Conclusion has been rewritten to better synthesize the study’s main outcomes, link findings to future applications (e.g., patient-derived sheets, drug testing platforms), and position this model as a step toward more physiologically relevant in vitro cancer systems.

Response to Reviewer #2

We sincerely thank Reviewer #2 for the positive and encouraging comments. We are pleased that the structure, clarity, and novelty of the manuscript were well received.

Finally, we have incorporated several improvements suggested by Reviewer #1, including expanded discussion to further strengthen the scientific depth and translational relevance of our findings.

Thank you

---

## [Decision Letter · Decision Letter 1]

PONE-D-25-04100R1Improving the robustness and efficiency of cell sheet protocol for breast cancer induction in animal model: a Geltrex™ plus Gelatin ApproachPLOS ONE

Dear Dr. Alshareeda,

Thank you for submitting your manuscript to PLOS ONE. After careful consideration, we feel that it has merit but does not fully meet PLOS ONE’s publication criteria as it currently stands. Therefore, we invite you to submit a revised version of the manuscript that addresses the points raised during the review process.

We look forward to receiving your revised manuscript.

Kind regards,

Mohamed Abdelkarim

Academic Editor

PLOS ONE

Journal Requirements:

Reviewers' comments:

Reviewer's Responses to Questions

**Comments to the Author**

1. If the authors have adequately addressed your comments raised in a previous round of review and you feel that this manuscript is now acceptable for publication, you may indicate that here to bypass the “Comments to the Author” section, enter your conflict of interest statement in the “Confidential to Editor” section, and submit your "Accept" recommendation.

Reviewer #1: (No Response)

2. Is the manuscript technically sound, and do the data support the conclusions?

Reviewer #1: Yes

3. Has the statistical analysis been performed appropriately and rigorously? 

Reviewer #1: Yes

4. Have the authors made all data underlying the findings in their manuscript fully available?

Reviewer #1: Yes

5. Is the manuscript presented in an intelligible fashion and written in standard English?

Reviewer #1: Yes

6. Review Comments to the Author

Reviewer #1: Thank you for the opportunity to re-evaluate the revised manuscript. The authors have made notable improvements following the major revision request. However, several key issues remain partially addressed: (1) no quantitative comparison of native versus modified cell sheet integrity; (2) reliance on crystal violet staining without complementary viability assays; (3) limited discussion on TNBC-specific molecular features, biomarkers, or targeted therapies; (4) unclear manuscript classification (research, methodology, or hybrid); and (5) no alternative scaffold materials suggested despite acknowledged cost and variability concerns. While these do not undermine the overall merit of the study, they are necessary for ensuring the manuscript’s scientific rigor and translational relevance. I therefore recommend minor revision, with the following considerations:

• Quantitative Analysis of Cell Sheet Integrity

o Issue: The manuscript lacks a quantitative comparison between native and modified cell sheets in terms of structural integrity.

o Recommendation (optional for authors): Please include relevant quantitative data if available, or provide a rationale for its exclusion. Reference to similar quantification methods in literature would strengthen this section.

• Viability Assays Beyond Crystal Violet

o Issue: Reliance on crystal violet staining limits interpretation of cell viability.

o Recommendation: Acknowledge this limitation and, if feasible, suggest alternative or complementary assays such as MTT, alamarBlue, or live/dead staining.

• Molecular and Translational Relevance to TNBC

o Issue: The discussion currently lacks depth on triple-negative breast cancer (TNBC)-specific molecular features or implications.

o Recommendation: Expand the discussion to include relevant TNBC biomarkers (e.g., EGFR, AR, PD-L1) and potential clinical applications of the methodology within this subtype.

• Clarification of Manuscript Type

o Issue: The manuscript's classification (original research, methodological paper, or hybrid) is ambiguous.

o Recommendation: Clearly state the article type in the abstract or introduction, ensuring alignment with the journal's submission categories.

• Discussion of Scaffold Material Alternatives

o Issue: While the authors acknowledge the cost and variability of current scaffold materials, no alternatives are discussed.

o Recommendation (optional): Briefly mention scalable or cost-effective alternatives (e.g., synthetic polymers, decellularized matrices) and potential for future optimization.

Please ensure these points are addressed or acknowledged in the revised version to further enhance clarity, reproducibility, and impact. I look forward to seeing the updated manuscript.

7. PLOS authors have the option to publish the peer review history of their article (what does this mean? ). If published, this will include your full peer review and any attached files.

**Do you want your identity to be public for this peer review?** For information about this choice, including consent withdrawal, please see our Privacy Policy .

Reviewer #1: No

---

## [Author Response · Author response to Decision Letter 2]

26 May 2025

We sincerely thank the reviewer for the thoughtful and detailed evaluation of our revised manuscript. We greatly appreciate the recognition of the improvements made thus far and the constructive suggestions to further enhance the scientific rigor and translational relevance of our work. Below, we address each point raised:

1. Quantitative Analysis of Cell Sheet Integrity

Comment: The manuscript lacks a quantitative comparison between native and modified cell sheets in terms of structural integrity.

Response:

At the stage of the research, the primary aim of this study was to provide a proof-of-concept for this approach. We appreciate the reviewer's suggestion to include a quantitative comparison. However, we would like to clarify that the MDA-MB-231 cell line used in our study known to have weak or absent cell junctions, which poses a significant challenge in assessing structural integrity using conventional quantitative methods in our model.

In vitro, MDA-MB-231 cells are typically cultured as a monolayer on a flat surface, such as a Petri dish, which allows for easy assessment of cell morphology and adhesion properties. However, when transitioning to a 3D environment, such as the layered gel system used in our study, the cells' behavior and interactions with the surrounding matrix change dramatically.

Moreover, the translation of cell sheets from an in vitro setting to an in vivo environment, such as animal tissue, introduces additional complexities, including changes in oxygen tension, nutrient availability, and mechanical stress. These factors can significantly affect the structural integrity of the cell sheet, making it difficult to design a quantitative assay that accurately reflects the cell sheet's behavior in both settings.

To evaluate the integrity of the cell sheets, we employed a complimentary approach. After building up our 3D model, we re-plated the cell sheets in a normal Petri dish to assess their ability to re-attach and grow. Notably, we found that the cells re-attached normally within 24 hours and continued to grow and proliferate over subsequent days. This is evident in Figure 2.C, where we used Crystal Violet staining after fixing the whole plate to demonstrate normal cell growth and morphology.

The fact that the cell sheets were able to re-attach and grow normally in a 2D environment after being cultured in the 3D Geltrex™ plus Gelatin system suggests that the modified cell sheet protocol does not compromise the intrinsic properties of the MDA-MB-231 cells. Instead, it appears to maintain their viability and ability to grow and proliferate.

While this approach may not provide a direct quantitative measurement of structural integrity, it offers a functional assessment of the cell sheets' ability to adapt to different environments and maintain their cellular properties. We believe that this complementary approach, combined with the qualitative observations of improved cell sheet morphology and stability in the modified system, provides a comprehensive understanding of the cell sheet's behavior and integrity in the context of our study.

Nevertheless, we have now included a rationale for the absence of such quantitative analysis in the revised manuscript (Results section, page 14, lines 327 – 334).

2. Viability Assays Beyond Crystal Violet

Comment: Reliance on crystal violet staining limits interpretation of cell viability.

Response: As clarified in comment 1, we appreciate the reviewer's comment. While Crystal Violet staining is a widely used method to evaluate cell growth and morphology, we acknowledge that it may not provide a comprehensive understanding of cell viability.

However, we would like to clarify that our use of Crystal Violet staining in Figure 2.C was intended to demonstrate the ability of the cell sheets to re-attach and grow in a 2D environment after being cultured in the 3D Geltrex™ plus Gelatin system, rather than to provide a definitive assessment of cell viability. The fact that the cells were able to re-attach and grow normally, as evidenced by the Crystal Violet staining, suggests that the modified cell sheet protocol does not compromise the intrinsic properties of the MDA-MB-231 cells.

Moreover, Crystal Violet staining is one of the methods to detect maintained adherence of cells, which is a critical aspect of cell sheet integrity. By using this method, we were able to assess the ability of the cell sheets to maintain their adherence properties after being cultured in the 3D system, and the results suggest that the modified protocol supports the maintenance of cell adherence. This is an important consideration, as cell adherence is a key factor in determining the overall integrity and functionality of the cell sheets.

3. Molecular and Translational Relevance to TNBC

Comment: The discussion currently lacks depth on triple-negative breast cancer (TNBC)-specific molecular features or implications.

Response: We have expanded the Discussion (page 22, lines 487 – 495) to include a more detailed overview of key TNBC-specific biomarkers, including EGFR, androgen receptor (AR), and PD-L1. At the same time we want to keep the focus on the main idea which is how our methodology could be leveraged in future targeted therapies or personalized treatment strategies for TNBC patients.

4. Clarification of Manuscript Type

Comment: The manuscript’s classification (original research, methodological paper, or hybrid) is ambiguous.

Response: Thank you for pointing this out. We have now clearly stated in both the Abstract and Introduction that this work should be considered “original research” presenting original research findings alongside methodological advancements. We have also ensured alignment with the journal’s submission categories (page 3, lines 57-58).

5. Discussion of Scaffold Material Alternatives

Comment: No alternative scaffold materials discussed despite acknowledged cost and variability concerns.

Response: We acknowledge the reviewer’s valuable point. In the revised Discussion (page 20-21, lines 454 – 457), we have briefly mentioned alternative scaffold options, including scalable synthetic polymers (e.g., polyethylene glycol [PEG]-based hydrogels) and decellularized matrices, which could be alternatives for future optimization and clinical translation.

We are grateful for the reviewer’s insightful feedback, which has helped us significantly strengthen the manuscript. We have carefully incorporated all suggestions and believe the revised version better addresses concerns regarding scientific rigor, reproducibility, and clinical relevance.

---

## [Editor Report · Decision Letter 2]

Improving the robustness and efficiency of cell sheet protocol for breast cancer induction in animal model: a Geltrex™ plus Gelatin Approach

PONE-D-25-04100R2

Dear Dr Alaa T. Alshareeda,

We’re pleased to inform you that your manuscript has been judged scientifically suitable for publication and will be formally accepted for publication once it meets all outstanding technical requirements.

Kind regards,

Mohamed Abdelkarim

Academic Editor

PLOS ONE

---

## [Editor Report · Acceptance letter]

PONE-D-25-04100R2

PLOS ONE

Dear Dr. Alshareeda,

I'm pleased to inform you that your manuscript has been deemed suitable for publication in PLOS ONE. Congratulations! Your manuscript is now being handed over to our production team.

Kind regards,

on behalf of

Dr. Mohamed Abdelkarim

Academic Editor

PLOS ONE